# Four Core Genotypes mice harbour a 3.2MB X-Y translocation that perturbs *Tlr7* dosage

Jasper Panten[1,2,8], Stefania Del Prete [1,8], James P. Cleland[1,3,8], Lauren M. Saunders[1], Job van Riet[1,4], Anja Schneider [1], Paul Ginno[1], Nina Schneider[1], Marie-Luise Koch[1], Xuqi Chen[5], Moritz Gerstung[4], Oliver Stegle [2,3], Arthur P. Arnold[5], James M. A. Turner [6], Edith Heard [3,7] ✉ & Duncan T. Odom [1] ✉

The Four Core Genotypes (FCG) is a mouse model system used to disentangle the function of sex chromosomes and hormones. We report that a copy of a 3.2 MB region of the X chromosome has translocated to the Y$^{Sry-}$ chromosome and thus increased the expression of X-linked genes including the single-stranded RNA sensor and autoimmune disease mediator *Tlr7*. This previously-unreported X-Y translocation complicates the interpretation of studies reliant on C57BL/6J FCG mice.

Sex-dependence of mammalian physiology and disease is shaped by both gonadal hormones (testicular vs ovarian) and sex chromosome complement (XX, XY or variations thereof). The "Four Core Genotypes" (FCG) mouse model was developed to decouple the relative contributions of sex hormones and chromosomes by moving the testis determinant *Sry* from the Y chromosome to an autosome[1]. The FCG model thus generates mice that are: XX with ovaries (XXO), XX with testes (XXT), XY with ovaries (XYO) and XY with testes (XYT) (Fig. 1a). The FCG mouse model has been used extensively in research articles (Supplementary Table 1)[2], affording insights into sex-biased immunity, cancer, Alzheimer's occurrence and obesity[3–7].

The popularity of the FCG model has recently surged in parallel with the increased attention on sex-biased and hormone-driven differences in health and disease. Here, we report that the very widely used line of C57BL/6J (B6) FCG mice has a previously-unreported copy of a multi-megabase region of the X chromosome fused to the Y chromosome with the *Sry* deletion (denoted Y$^{Sry-}$), thus substantially increasing the expression level of several X-linked genes including *Tlr7*, a single-stranded RNA sensor causally related to autoimmune disease pathogenesis[8,9].

## Results and discussion

While analysing single-cell RNAseq data from the spleen and liver of adult B6 FCG mice (Supplementary Figs. 1–4), we noticed that several genes located on a single contiguous region of the X chromosome adjacent to the pseudoautosomal region (PAR) (Fig. 1b) are approximately 2-fold upregulated in XYT and XYO cells compared with their XX counterparts (Fig. 1c–h). This observation was unexpected because X-inactivation should ensure equal X-linked gene dosage between XX and XY genotypes. In addition, these genes showed approximately 2-fold higher expression in FCG XYT liver cells than in wild-type male (XYT-WT) cells (Supplementary Fig. 5). Based on these results, we hypothesised that a segment of X has translocated to the Y$^{Sry-}$. Fusions with other chromosomes are unlikely to have survived the extensive backcrossing used to move the FCG model onto the C57BL/6J (hereafter referred to as B6) background[8].

The identified set of nine X-linked genes showed cell type-specificity in their level of expression. Five of the nine genes (*Arhgap6*, *Msl3*, *Frmpd4*, *Prps2* and *Tmsb4x*) are detectably expressed in splenocytes, hepatocytes and Kupffer cells; however, *Hccs* and *Amelx* are not expressed in Kupffer cells and *Tlr7* and *Tlr8* are not expressed in hepatocytes (Fig. 1c–h and Supplementary Figs. 6–7). Beyond the

[1]Division of Regulatory Genomics and Cancer Evolution, German Cancer Research Center (DKFZ), Heidelberg, Germany. [2]Division of Computational Genomics and Systems Genetics, German Cancer Research Center (DKFZ), Heidelberg, Germany. [3]European Molecular Biology Laboratory (EMBL), Heidelberg, Germany. [4]Division of Artificial Intelligence in Oncology, German Cancer Research Center (DKFZ), Heidelberg, Germany. [5]Department of Integrative Biology & Physiology, University of California, Los Angeles, USA. [6]Sex Chromosome Biology Laboratory, The Francis Crick Institute, London, UK. [7]Collège de France, Paris, France. [8]These authors contributed equally: Jasper Panten, Stefania Del Prete, James P. Cleland. ✉e-mail: edith.heard@embl.org; d.odom@dkfz.de

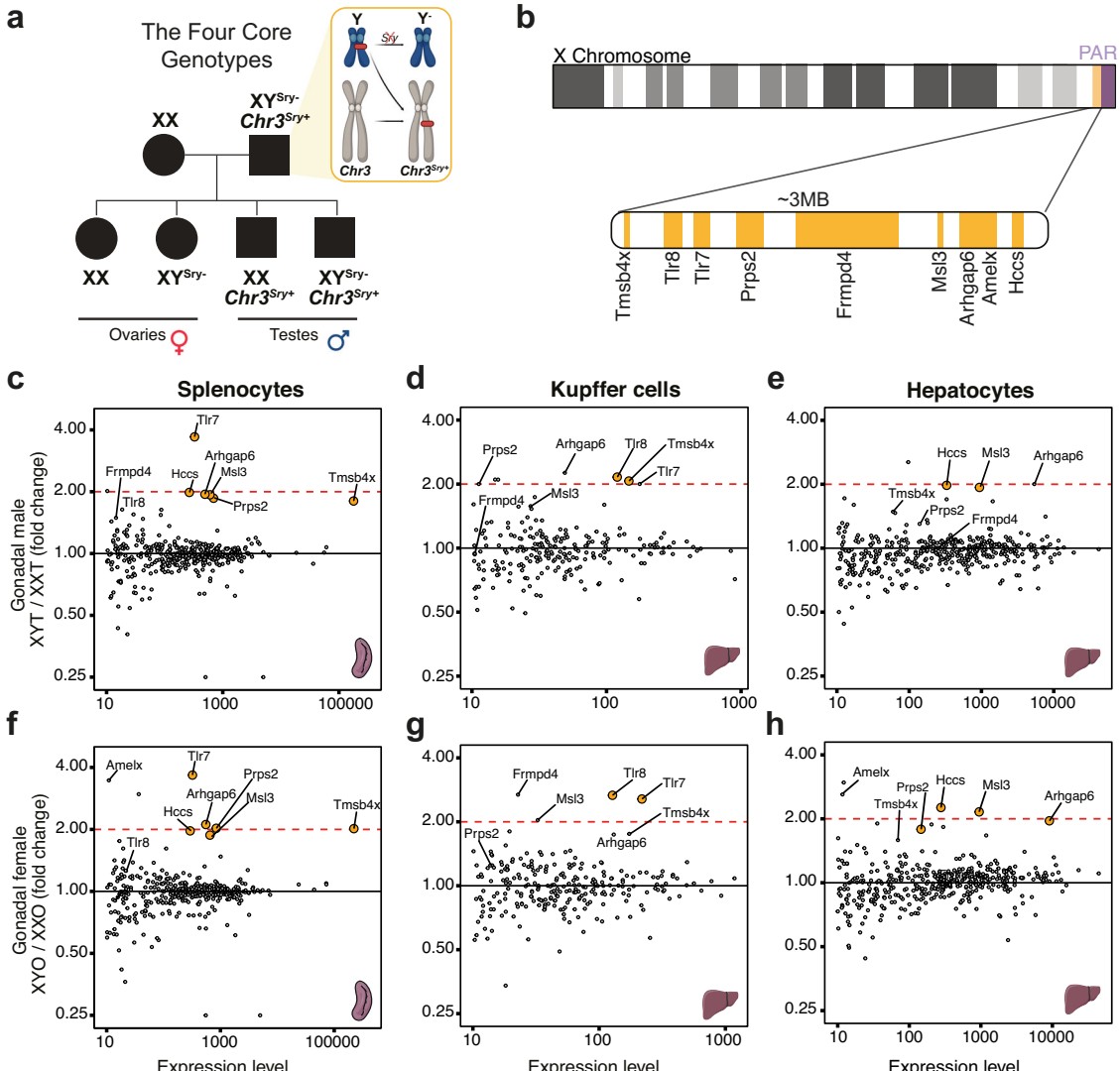

**Fig. 1 | Transcriptional dosage of genes within a PAR-adjacent 3.2 MB region of the X chromosome is doubled in C57BL/6 J FCG XY mice. a** Schematic representation of the FCG model. **b** A 3.2 MB region is flanking the PAR of the X chromosome, containing nine genes. **c, f** Differential expression analysis of X chromosome genes on pseudo-bulked single-cell RNA sequencing from all spleen cells between (**c**) XYT and XXT mice and between (**f**) XYO and XXO mice. **d**–**h** Differential expression analysis of X chromosome genes on pseudo-bulked single-nucleus RNA sequencing of liver (**d**–**g**) Kupffer cells and (**e**–**h**) hepatocytes isolated from XYT and XXT mice. The *x*-axis indicates the expression level as total read counts. Highlighted in orange are all genes in the PAR-adjacent region that have a positive fold change and an adjusted *p*-value < 0.1 (DESeq2 Wald test). The chromosome, spleen and liver icons were created using BioRender.com. Icons in the figure panels (1**a**, 1**c**–**h**) were created with BioRender.com and are released under a CC-BY-NC-ND license.

spleen and liver, the Tabula Muris dataset indicates that the nine X-linked genes are widely but unevenly expressed across cell types (Supplementary Table 2)[9]. Despite the higher expression of the nine putatively-translocated genes (Fig. 1c–h), we observed relatively mild effects on the expression of other X-linked genes (Fig. 1c–h) and autosomal genes (Supplementary Fig. 6).

We then directly identified the amplified region via whole genome sequencing of the two B6 FCG founder XYT mice obtained commercially, alongside a B6 wild-type male with paired-end short-read sequencing (Fig. 2a, b and Supplementary Figs. 8–10). This revealed that the amplification is a clean duplication of the B6 sequence spanning 3.2 MB, adjacent to the PAR of the X chromosome. We further resolved the bounds of the amplified region by whole-genome sequencing of FCG XYT and wild-type male mice on an F1 B6-FCG x CAST/EiJ (CAST) background, where the duplicated X-linked region can be genetically identified as originating from both the maternal CAST X chromosome and the paternal B6 genome (Fig. 2c, d and

Supplementary Fig. 11). This demonstrates that the duplication is paternally inherited and likely fused to the Y chromosome. We identified the amplification as an approximately 3.220.000 base pair region mapping to chrX: 165.530.000 − 168.750.000 on *GRCm39*, containing nine annotated genes (*Tmsb4x, Tlr8, Tlr7, Prps2, Frmpd4, Msl3, Arhgap6, Amelx and Hccs*), of which six are detectably upregulated in an FCG XY-biased manner in the cells we profiled.

To verify whether the identified X amplification has indeed translocated to the Y, we performed DNA FISH detection and used probes that target the whole Y chromosome and a -200 kb domain located in the middle of the X chromosome amplification, with experiments performed in wild-type male and B6 FCG XYT primary splenocytes (Fig. 2e). We observe that 92% of FCG XYT cells harbour two X domains instead of one (Fig. 2f) and that one of these two X domains is in close proximity to the Y chromosome (Fig. 2g). Finally, to identify the precise translocation site on the Y, we performed long read sequencing and Targeted Locus Amplification (TLA). Nanopore

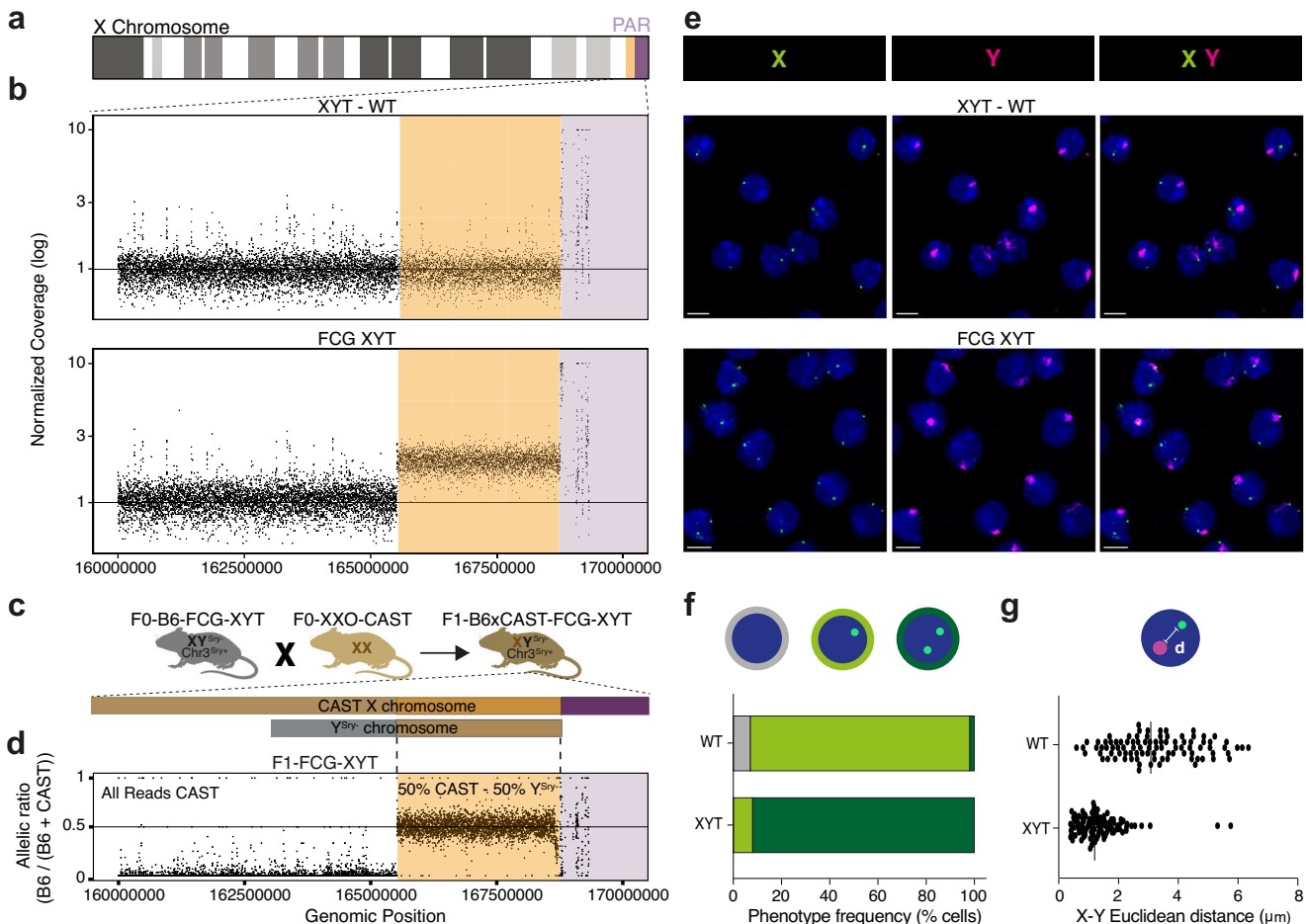

**Fig. 2 | Whole genome sequencing and DNA FISH reveal an X-Y translocation in C57BL/6J FCG XY mice. a** Overview of the X-chromosome, where orange indicates the location of the amplified region, and purple indicates the PAR. **b** Coverage of sequencing reads normalised to the median autosomal coverage in 1 kb windows of a C57BL/6J wild-type male (top) and the FCG XYT founder mouse (bottom). **c** Schematic of the F1 cross between the XYT-B6 male mice and the XXO-CAST female mice. The sex chromosomes in F1 mice are shown, the $Y^{Sry-}$ chromosome holds the duplication. **d** Ratio of haplotype-resolved coverages in an F1 FCG XYT hybrid. The presence of B6-specific reads in the amplified region reveals a paternally inherited 3.2 MB region duplicated in the B6 FCG model. **e** DNA FISH of the whole Y chromosome (magenta) and a 200 kb domain of the X amplification

(green) in C57BL/6J wild-type male (top row) and B6 FCG XYT (bottom row) primary splenocytes. Images represent maximum intensity projections. Scale bars represent 5 μm. **f, g** DNA FISH of the whole Y chromosome (magenta) and a ~200 kb domain, centred on *Msl3* and located in the middle of the X chromosome amplification (green). Quantification of X domain number (**f**) and Euclidean distance between the Y domain and its nearest X domain (**g**). *n* = 100 cells across five fields of view per genotype. Vertical lines represent the median. Error bars represent standard deviation (**g**). Source data are provided as a Source Data file. The mouse icons were created with BioRender.com. Icons in the figure panel (**2c**) was created with BioRender.com and is released under a CC-BY-NC-ND license.

sequencing revealed the translocation boundary to overlap with a LINE repeat sequence (L1MdTf_III family, Supplementary Fig. 12). TLA and Nanopore analysis shows that the amplified chromosome X sequence integrated around chrY: 90.210.000 (Supplementary Fig. 13). Thus, multiple lines of evidence support that the increased expression observed in B6 FCG XY genotype mice is caused by translocation of a 3.2 MB segment of the X to the $Y^{Sry-}$.

Our discovery that a copied 3.2 MB segment of the X has become fused to the $Y^{Sry-}$ chromosome reflects the long-known proclivity for regions adjacent to the PAR to undergo genetic exchange[10]. Indeed, a similar translocation has occurred in the YAA model, wherein a 4 MB stretch of the X chromosome is fused to the Y chromosome, with consequences to gene expression and susceptibility to auto-immunity disease[11]. Translocations such as these may occur stochastically, as no comparable one has occurred in the decades-old MF1 lineage of FCG mice (Supplementary Figs. 14, 15).

While we purchased our B6 FCG XYT colony founders in the summer of 2022, the model has been commercially available following

extensive backcrossing to move the FCG model onto a C57BL/6J background, completed in 2010[8]. Upregulation of the translocation genes we identified in FCG XY mice has been previously reported[12–15], suggesting that this genetic alteration has been circulating at least since 2010. Whole genome sequencing of B6 FCG XYT samples dating back to 2010 confirmed the presence of the translocation (Supplementary Fig. 14). However, we have also identified an independently backcrossed B6 FCG colony without the translocation (Supplementary Fig. 14), which is available to the scientific community via The Jackson Laboratory.

Given that the nine translocated genes encode important proteins that are widely expressed and have been reported to alter core cellular functions when overexpressed (Supplementary Table 2), the translocation could potentially impact prior studies and ongoing projects that rely on the currently available B6 FCG model. The presence of the translocation indicates that any differences between B6 FCG XX and XY mice with the same type of gonads could be attributed to the number and type of sex chromosomes. However, these differences

might also result from the extra dosage of one or more genes within the translocated region. Further research is needed to distinguish between these two possibilities.

Our data suggests that genetic models perturbing sex chromosome complement should be routinely genome-sequenced, due to inherent genetic instability of PAR-adjacent regions[10,11,16]. Meanwhile, the FCG mouse model carrying an X-Y translocation can be reliably used to identify gonadal hormone contributions by comparing the XXO and XXT groups. Our findings emphasise the importance of cross-validating results arising from the FCG model using other sex chromosome complement models and wild-type males, as has long been recommended[7].

## Methods

### Mouse colony management
Four Core Genotypes colony founders on the C57BL/6 J background were obtained from The Jackson Laboratory (strain 010905)[8]. The Y$^{Sry}$-chromosome present in the C57BL/6 J FCG is on a 129S background. Two XYT individuals were mated with wild-type C57BL/6 J (Janvier-Labs) or CAST/EiJ (bred at the German Cancer Research Centre) females to respectively generate pure C57BL/6 J and F1 B6 x CAST hybrid XXO, XXT, XYO and XYT progeny. Wild-type C57BL/6 J males (Janvier-Labs) were imported and housed for one week prior to sacrifice. All animals were maintained as virgins (apart from the two B6 FCG XYT founders) and housed in groups of up to six mice in Tecniplast GM500 IVC cages with a 12 h light/dark cycle. Mice had ad libitum access to water, food (Kliba 3437) and environmental enrichment. The colony was regularly controlled for infections using sentinel mice to ensure a healthy status. All breedings and organ collections were carried out with the approval of the Regierungspräsidium Karlsruhe. Animal genotype (i.e., XXO, XXT, XYO or XYT) was assigned based on an independent assessment of sex chromosome complement and gonadal sex. Sex chromosome complement was initially determined by PCR detection of the Y chromosome using published primers YMTFP1 (5′ CTG GAG CTC TAC AGT GAT GA 3′) and YMTRP1 (5′ CAG TTA CCA ATC AAC ACA TCA 3′)[17] and confirmed by detection of *Xist* mRNA (specific to the XX genotypes) and *Sry* (specific to the XY genotypes) sequencing signal. Gonadal sex was determined by visual assessment of external morphology, initially at the time of weaning and confirmed at the time of sacrifice.

### Tissue collection and processing
Animals were sacrificed by carbon dioxide inhalation at 12 weeks of age unless otherwise stated. Whole livers were dissected out, transferred to 5 ml tubes, snap-frozen on liquid nitrogen and stored at −80 °C. Frozen livers were pulverised with the cryoPREP® Automated Dry Pulveriser (Covaris #CP02) using the TT2 holder, TT2 consumables and impact setting 6 and pulverised material stored at −80 °C. Whole spleens were dissected, transferred to a petri dish containing DMEM (Gibco # 41966029) with 5% foetal bovine serum (FBS, Thermo Fisher # 16140071) and processed directly for single-cell isolation. All tissue collections were performed between 8:00 and 12:00 to approximately control for circadian regulation of organ physiology.

### Single-cell RNA sequencing: sample preparation, library preparation and sequencing
Dissected spleen fragments were transferred to a 40-µm cell strainer (Greiner # 542040) and ground with a syringe plunger. The cell suspension was strained twice. Strainers were washed with additional DMEM (Gibco # 41966029) with 5% FBS, and cells were pelleted by centrifugation at $300 \times g$ for 4 min at 4 °C. Cell pellets were carefully resuspended in 300 µL of ACK lysis buffer (Lonza # BP10-548E) for 1 min in ice. After incubation, cells were washed with 1 mL of DMEM with 5% FBS and pelleted. Cells concentration was determined using 0.4% trypan blue stain (Invitrogen) and Countess II automated cell

counter (Invitrogen). $1 \times 10^7$ cells were processed for dead cell depletion using the Dead Cell Removal Kit (Miltenyi Biotec # 130-090-101) as per the manufacturer's instructions. The magnetically labelled dead cells were retained using MS MACS Columns (Miltenyi Biotec # 130-042-201) on a MiniMACS Separator (Miltenyi Biotec # 130-042-102). The live cell fraction was pelleted and resuspended in 1X PBS with 0.04% bovine serum albumin (BSA, Miltenyi Biotec # 130-091-376) to a final concentration of $1 \times 10^6$ cells/mL (1000 cells per µL).

Single-cell RNA was performed with 20.000 cells which were encapsulated into droplets in the Chromium Controller instrument using Chromium Next GEM Single Cell 3′ Reagents Kits v3.1, according to the manufacturer's recommendations. Briefly, single-cell lysis and barcoded reverse transcription (RT) of mRNA happened into the droplets. Twelve cycles were used for cDNA amplification and sample index PCR to generate Illumina libraries. Quantification of the libraries was carried out using the Qubit dsDNA HS Assay Kit (Life Technologies), and cDNA integrity was assessed using D1000 ScreenTapes (Agilent Technologies). Libraries were pooled in an equimolar manner, and 1000pM were paired-end sequenced on an Illumina NextSeq2000 (R1 = 30 cycles, R2 = 8 cycles, I1 = 0 cycles, I2 = 199 cycles).

### Single-cell RNA sequencing: data processing and analysis
Genomic references for C57BL/6 (GRCm38) were generated using CellRanger *mkref* (v7.0) using sequence and gene annotations from Ensembl (release 94). For mapping of C57B6 / CAST/EiJ F1 mouse data, a joint reference was constructed based on the GRCm38 reference in which Single Nucleotide Polymorphism (SNP) positions between mm10 and CAST/EiJ were N-masked. These SNPs were derived from (Keane2011, ftp://ftp-mouse.sanger.ac.uk/current_snps/mgp.v5.merged.snps_all.dbSNP142.vcf.gz). Filtered count matrices were generated using CellRanger *count* (v6.1.1) using default settings, which include reads mapped to introns. Low-level analysis of scRNA-Seq data was then performed, largely using functions from the *scran* (v1.24.1) and *scater* (v1.20.1) R packages[18,19]. First, cells with less than 500 UMIs and 500 detected genes were removed. Next, counts were normalised using the *computeSumFactors* function and log-transformed. For cell type annotation, we used mutual nearest neighbour-based batch correction using the function *MNNcorrect* with the library as the batch variable to exclude species-specific and technical variation across samples[20]. The resulting corrected matrix was used for dimensionality reduction by principal component analysis (*prcomp, stats*), tSNE (*Rtsne, Rtsne*, v0.15) and UMAP (*umap, umap*, v0.2.7.0). To identify cell-type clusters, we used graph-based community detection using the Louvain algorithm implemented by the functions *buildSNNGraph* and *cluster_louvain* of the package *igraph* (v1.2.10). Cell type labels were defined by label transfer from the ImmGenData dataset (celldex R package, v1.6.0) using the SingleR R package (v1.10.0). Doublets were identified and excluded using the scDoubletFinder package (v1.15.4).

### Single-nucleus RNA sequencing: sample preparation, library preparation and sequencing
Nuclei isolation was performed as described previously[21], with the following modifications. ~20 mg pulverised frozen liver tissue was dounced ten times with pestle A and ten times with pestle B (Sigma # 8938) in 1 ml hypotonic lysis buffer solution B containing 10 mg/ml BSA (Sigma # 126609). The homogenate was then diluted 1:5 in fresh lysis buffer, incubated on ice for 10 min, filtered through a 30 µm CellTrics filter (Sysmex # 04-004-2326) and spun at 700 rcf for 5 min. Pelleted nuclei were resuspended in 1 ml SPBSTM supplemented with 1% DEPC, spun again and resuspended in 1 ml SPBSTM. Nuclei were fixed by the addition of 4.3 ml pre-chilled fixation solution (4 ml MeOH and 300ul 16.7 mg/ml DTSSP, Sigma # 803200) and incubation on ice for 15 min, rehydrated by addition of 3 volumes of SPBSTM, spun at 700 rcf for 10 min, resuspended in 1 ml SPBSTM and stored at −80 in

250 μl aliquots. Nuclei quality and concentration were quantified using a LUNA-FX7 automated cell counter (Logos). Library preparation was performed as described previously[21], with the following modifications. Nuclei were passed through a 20 μm EasyStrainer filter (Greiner # 542120) prior to distribution across the reverse transcription plate. Following protease digestion, the optimal Tn5 concentration was empirically determined by testing a 2-fold dilution series (0.25, 0.13, 0.06, 0.03 N7-loaded Tn5 per 5 μl reaction) on a small subset of wells (2 per dilution). Final libraries were subjected to a two-sided (0.45x/0.80x) AmpureXP bead cleanup but not gel extraction. Libraries were sequenced on a NovaSeq 6000 system (Illumina) with a S4 200 cycles reagent kit (R1 = 34 cycles, R2 = 184 cycles, I1 = 10 cycles, I2 = 10 cycles).

### Single-nucleus RNA sequencing: data processing and analysis

**Sample demultiplexing and alignment.** For handling the demultiplexing and subsequent downstream alignment and read-counting of sci-RNA-seq3 data, we designed a custom *Snakemake*[22] workflow termed sci-rocket, which has been made publicly-available under the MIT licence. (https://github.com/odomlab2/sci-rocket). Raw sequencing data (binary base calls) were converted to paired-end read sequences using *bcl2fastq* (v2.20.0.422) with the PCR index #1 and PCR index #2 sequences added within the read-name using the following parameters: bcl2fastq -R < bcl > --sample-sheet < samplesheet > --output-dir < out > --loading-threads < n > --processing-threads < n > --writing-threads < n > --barcode-mismatches 1 --ignore-missing-positions --ignore-missing-controls, --ignore-missing-filter --ignore-missing-bcls --no-lane-splitting --minimum-trimmed-read-length 15 --mask-short-adaptor-reads 15

Subsequently, raw paired-end sequences were split into smaller, evenly-sized chunks (n = 75) using *fastqsplitter* (v1.2.0), which were analysed in parallel during the subsequent sample-demultiplexing procedure. Reads were assigned to their respective sample based on the combinatorial presence of four barcodes within the R1 read; PCR index 81 (p5), PCR index 82 (p7), ligation primer and the reverse transcriptase (RT) primer used during the sci-RNA-seq3 protocol. Each barcode was 10 nucleotides in length, except for the ligation barcode, which could be either 9 or 10 nucleotides in length. From each R1 read, these barcodes were retrieved and matched, with a hamming distance of max. 1nt, to the white list of barcode sequences which could be present within the sequencing run. If all four barcodes could be matched, each paired-end read was deposited into their respective sample-specific.fastq files. The read containing the combinatorial barcode (R1) was modified to only contain the sequence of the matching white-listed barcodes (if hamming distance > 0) and UMI of 8nt in length. Ligation barcodes of 9 nucleotides in length were padded with an extra G in order to always produce sequences of 48 nucleotides in length; PCR Index 11 (10nt), PCR index 42 (10nt), ligation (10nt), RT (10nt) and UMI (8nt). The ligation white list was also adjusted accordingly. Read-pairs without all four barcodes matching the white lists were discarded into a separate file containing all discarded reads with information on which barcode sequence(s) were non-matching. Demultiplexed.fastq files from each parallel job (n = 75) were merged to produce a single sample-specific paired-end.fastq file (R1 and R2).

After sample-demultiplexing, remaining adaptor sequences and low-quality bases (≥ Q15) were trimmed using *fastp* (v0.23.4)[23]. Read-pairs with mates containing fewer than 10 nucleotides after trimming were discarded. Alignment of trimmed reads was performed against GRCm39 (M31) with GENCODE (v31) annotations[24] using STARSolo (v2.7.10b)[25] with the following parameters:

STAR --genomeDir < index > --runThreadN < n > --readFilesIn < R2 > < R1 > --readFilesCommand zcat --soloType CBUMI_Complex --soloCBmatchWLtype Exact --soloCBposition 0_0_0_9 0_10_0_19 0_20_0_29 0_30_0_39 --soloUMIposition 0_40_0_47 --soloCBwhitelist < wl_p7 > < wl_p5 > < wl_lig > < wt_rt > --soloCellFilter CellRanger2.2

< n_expected_cells > 0.99 10 --soloFeatures GeneFull --soloCellReadStats Standard --soloMultiMappers EM --outSAMtype BAM SortedByCoordinate --outSAMunmapped Within --outFileNamePrefix < sample > --outSAMmultNmax 1 --outSAMstrandField intronMotif --outFilterScoreMinOverLread 0.33 --outFilterMatchNminOverLread 0.33 --outSAMattributes NH HI AS nM NM MD jM jI MC ch XS CR UR GX GN sM

In addition, indexes of BAM files were generated using sambamba (v1.0)[26]. Briefly, this generated sparse matrices containing the UMI counts for each gene per cellular barcode. For each gene, we counted the total set of UMI of all intronic, exonic, and UTR-overlapping reads based on GENCODE (v31) annotation. Reads mapping to multiple genes (multi-mappers) were counted using the Expectation-Maximisation (EM) algorithm available within STARSolo. After visual inspection of the sample-wise knee plots with the STARSolo-designated UMI threshold to distinguish ambient RNA from the true cell, we opted to adopt these UMI thresholds as-is.

**Data import and quality control.** Using monocle3 (v1.3.1)[27], we combined the sample-wise UMI matrices generated by STARSolo and imported all protein-coding, lncRNA and immunoglobulin genes and removed predicted-only genes for a total of 25,064 genes across 131,975 cells. Next, we determined the number of relevant principal components capturing at least ≥ 1% of the variance (n = 20) and used these principal components in subsequent dimension reduction using the default monocle3 workflow whilst using the number of UMI per cell to regress out potential batch-effects. Initial major cell-type clusters were identified by Leiden clustering ($k = 20$, resolution $= 1^{(e-6)}$)[28]. To detect potential doublets, we performed scDblFinder (v1.14.0)[29] to estimate captured cells which resembled artificial combination(s) of multiple major clusters (n = 16) within our experiment. We utilised all relevant principal components, capturing at least ≥ 1% of the variance (n = 20), assuming a maximum doublet rate of max. 2% ($dbr = 0.02$) and used the default number of artificially generated doublets. In addition, cells with a total UMI count $\geq \mu * 3\sigma$ (based on all cells) were flagged as being potential outliers (n = 4526). After visual inspection of cells flagged as doublet or UMI-outlier (n = 4626; 1.1% of total cells) within the UMAP space, we removed these cells prior to downstream analysis.

### Differential expression testing

Differential expression analysis of single-cell and single-nucleus RNA-Seq data was performed as a pseud-bulk per individual mouse and using the DESeq2 package[30]. To this end, raw read counts for each library were summed up, and genes with fewer than 10 reads per sample were excluded as lowly expressed. Then DESeq2 (v1.36.0) was used to compute size factors (*estimateSizeFactors* function) and to detect differentially expressed genes between XY and XX genotypes with default parameters (*DESeq* function), separately for the two gonadal groups. Genes were considered differentially expressed at an FDR of 10% (Benjamini-Hochberg correction). To decrease the multiple testing burden, we only test genes on the X-chromosome in this analysis. The single-nucleus RNA-Seq dataset was analysed in the same way, but separating individual cell types.

### Whole genome sequencing

DNA was isolated from ~30 mg frozen liver and 2mm³ ear punch biopsies using a protocol incorporating elements of the AllPrep DNA/RNA Micro Kit (Qiagen) and a DNeasy Blood and Tissue Kit (Qiagen). Frozen tissue samples were homogenised by adding 600 μL RLT containing 1% β-Mercaptoethanol and a 5 mm stainless steel bead (Qiagen # 69989) and processing for 40 s at 15 Hz with the TissueLyser II. Lysed samples were supplemented with 20 μl proteinase K (from the Blood and Tissue Kit) and incubated at 56 °C for 10 min. Half of the digested sample was combined with 500uL Buffer AL (Qiagen Blood & Tissue), mixed by vortexing and incubated at 56 °C and 500 rpm for 30 min. An

equal volume of absolute ethanol was added to the samples and mixed thoroughly. The precipitated mixture was transferred to a DNeasy Mini column and washed twice with 500 μl buffer AW1 and subsequently with 500 μl AW2. After drying the membrane for 2 min at 20,000 × $g$, DNA was eluted using 50 μl AE buffer pre-warmed to 70 °C. DNA quality and quantity were assessed using a Genomic DNA ScreenTape (Agilent Technologies) and Qubit dsDNA BR Assay Kit (Life Technologies).

Dual Index whole genome libraries were prepared with 300 ng of genomic DNA using the NEBNext Ultra II kit (E7805S/L). Fragmentation time was optimised and occurred at 37 °C for 20 min. Adaptor-ligated DNA was size selected with AMPure XP Beads (Beckman Coulter) to select an insert size of approximately 150–200 bp. Libraries were amplified 6 cycles using the NEBNext UDI primers (E6440). Library size and molarity were determined using a D5000 ScreenTape (Agilent Technologies) and Qubit dsDNA HS Assay Kit (Life Technologies). The final library size was approximately 600–700 bp. Libraries were pooled in an equimolar manner, and 750pM was sequenced, paired-end 150 bp, using the NextSeq 2000 platforms (300 cycle P3 chemistry) to aim for 200Mio reads per sample.

## Whole-genome sequencing: data processing and analysis
For the whole genome sequencing data, reads were first trimmed using Trim Galore (v0.6.10) and cutadapt (https://github.com/FelixKrueger/TrimGalore) using the options trim_galore --paired --three_prime_clip_R1 1 --three_prime_clip_R2 1 --fastqc --cores 8 --stringency 3. Next, reads were aligned to a GRCm39 in which all SNP positions between C57B6 and CAST/EiJ strains are N-masked (see also Single-cell RNA sequencing: data processing and analysis) using bowtie2 (v2.3.5.1) with default options[31]. PCR duplicates were deduplicated when they mapped to the exact same positions using the samtools (v1.15.1) programs collate, fixmate, sort and markup. Finally, for the F1 mouse samples, reads mapping to the C57B6 or CAST/EiJ genomes were assigned to either parental haplotype (or unassigned) using SNPsplit (v0.5.0)[32]. From the sorted and deduplicated bam-files, both total and allele-specific coverage tracks with windowsizes 1 kb and 1000 kb were generated using the *bamProfile* function from the *bamsignals* package (v1.28.0).

## Nanopore Sequencing
DNA was isolated from ~5 mg frozen liver using Puregene DNA Kit (Qiagen) with the following modification. Proteinase K was added to the sample, which was incubated at 55 °C until fully dissolved. DNA size and molarity were determined using a gDNA ScreenTape (Agilent Technologies) and Qubit dsDNA BR Assay Kit (Life Technologies). The High Molecular Weight DNA fragment length was approximately 60Kb. ~20ug of DNA in 50uL were sheared in a Covaris g-Tube for 1 min at 5000 rpm to aim for 30 Kb fragment size. Libraries were prepared using the Ligation Sequencing DNA V14 (SQK-LSK114) Kit (Nanopore Oxford) as per the manufacturer's instructions, and 20fmol was sequenced using the PromethION machine.

## Nanopore sequencing: data processing and analysis
Raw nanopore sequencing data was converted to fastq files. The resulting reads were mapped to the GRCm39 genome using minimap2 (v2.24) using the options -t 8 -a -x map-ont and visualised using the Integrative Genome Viewer (v2.12.3).

## Targeted locus amplification (TLA)
Ten million primary splenocytes were isolated from five-month-old XYT-WT and XYT mice as described above for scRNA-seq and shipped on dry ice to Cergentis (Utretcht, NL), who performed the TLA procedure[33]. First, DNA was crosslinked, fragmented, ligated, de-crosslinked and shuffled to get circular templates. Second, inverse PCR was performed using independent primer sets respectively targeting

the 5' end of the chrX duplication (forward TTCAGTAGGTGTGACA-GAGA; reverse ACATAATTATAGCGCTGGCA), 3' end of the chrX duplication (forward GCATCTGTCTGCTTACTTTG; reverse GGTC TGGATACCTTTCAAGG) and the chrY translocation site identified with the first two pairs (forward TAAAAGGGGAGGTAGACACA; reverse GTGTTCAAACCAGACTCCT). Third, PCR products were purified, library prepped using the Nextera DNA Flex (Illumina) chemistry, pooled and sequenced on an Illumina system (151 bp paired-end, ~1mio reads per PCR product). Fourth, sequencing reads were aligned to the GRCm38/mm10 mouse genome, and breakpoint sites were detected based on coverage peaks and fusion-reads.

## DNA fluorescence in situ hybridisation (FISH)
Primary splenocytes were isolated from one five-month-old XYT-WT and one XYT mouse as described above for scRNA-seq. 10 million splenocytes from each mouse were spun at 200 rcf and room temperature (RT) for 5 min, resuspended in 3:1 methanol:acetic acid fixative, incubated at −20 °C for at least one hour, spun at 200 rcf and RT for 5 min, resuspended in 45% acetic acid, incubated at RT for 5 min, spotted onto SuperFrost Plus Gold microscope slides, air dried overnight at RT and stored at 4 °C until use. Labelled *Msl3* probes were generated using a nick translation kit (Abbott) and midi-prepped bacterial artificial chromosome RP23-391N16 (BACPAC Genomics) as a template. FISH was performed largely according to the instructions provided by the nick translation kit manufacturer (https://www.molecular.abbott/int/en/vysis-fish-knowledge-centre/nick-translation-kit-preparing-the-reagents) with the following modifications. Cells were incubated in 0.1 mg/ml RNase A and 10 μ/ml RNase H diluted in 2x SSC for 1 h at 37 °C[34] prior to hybridisation. Hybridisation was performed for 21 h using 5ul Msl3 probes resuspended in Y chromosome paint solution (MetaSystems Probes #D-1421-050-OR) and 16 mm diameter coverglass. Samples were counterstained with 5 μg/ml Hoechst in 2x SSC for 5 min at RT, mounted with Prolong Diamond mounting medium and allowed to cure at RT for at least 24 h. Images (7 μm Z-stacks with 1 μm step size) were acquired with an IXplore Spin spinning disc confocal microscope equipped with a 100x oil immersion objective (Olympus). Five non-overlapping fields of view (FOV) were acquired per genotype. Images were quantified in Fiji using the Cell Counter plugin. For each genotype, 100 cells across five FOVs were counted. Specifically, cells with a clear Y domain were selected blind to the X signal, then X domain number and X-Y distance (i.e., distance from the centre of the Y domain to its nearest X domain) were quantified.

## Reporting summary
Further information on research design is available in the Nature Portfolio Reporting Summary linked to this article.

# Data availability
The whole genome sequencing data has been submitted to ArrayExpress under the accession number E-MTAB-13585. The spleen single-cell RNA-sequencing data has been submitted to ArrayExpress under the accession number E-MTAB-13586. The liver single-nucleus RNA-sequencing data has been submitted to ArrayExpress under the accession number E-MTAB-13700. The spleen TLA data has been submitted to ArrayExpress under the accession number E-MTAB-14096. The long-read whole genome sequencing data has been submitted to ArrayExpress under the accession number E-MTAB-14291. Raw confocal microscopy data has been submitted to the EMBL-EBI BioImage Archive and is available under accession number S-BIAD1282. DNA FISH quantification data are provided in this paper as Source data. Source data are provided in this paper.

# Code availability
Our custom *Snakemake* workflow (sci-rocket) used in processing the sci-RNA-seq3 data has been made publicly available under the MIT

licence (https://github.com/odomlab2/sci-rocket). Additional custom code used for the processing, analysis and visualisation of data supporting this manuscript is available via GitHub (https://github.com/odomlab2/fourcore_transloc).

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

## Acknowledgements

We thank all members of the Odom and Heard labs (in particular Dr. Tim Pollex and Dr. Agnese Loda); the DKFZ Central Animal Laboratory and Next Generation Sequencing core facilities (in particular Nina Glaser); the EMBL Advanced Light Microscopy Facility; and Cergentis for performing the TLA. This research was financially supported by core funding from the Helmholtz Association (to D.T.O. and O.S.) and EMBL (to E.H. and O.S.), a grant from the European Research Council (788937 to D.T.O.), a grant from the Wilhelm Sander Stiftung (to D.T.O. and E.H.), a DKFZ Postdoctoral Fellowship (to J.P.C.), state funds approved by the State Parliament of Baden-Württemberg for the Innovation Campus Health + Life Science Alliance Heidelberg Mannheim (to JvR) and a grant from the Bundesministerium für Bildung und Forschung Germany, project MERGE, Förderkennzeichen 031L0174C (to O.S.). A.P.A. and X.C. funding streams include US NIH grants MH59268, NS045966, NS043196, HL131182, and DK083561. Icons in the figure panels 1a, 1c–h, 2c Supplementary Fig. 1a–h, 2a–h, 3a–f, 4a, b, 6a–f and 7 were created with BioRender.com and are released under a CC-BY-NC-ND license.

## Author contributions

J.P. and S.D.P. made the initial discovery. S.D.P. performed scRNA-seq experiments with assistance from N.S., J.P.C. and L.M.S. performed snRNA-seq with assistance from A.S. and M.L.K., S.D.P. performed whole-genome sequencing with assistance from A.S. and X.C., J.P.C. performed FISH experiments. J.P., S.D.P., L.M.S. and J.v.R. performed computational analysis of sequencing data with assistance from P.G., J.P.C. coordinated and assisted with TLA experiments. S.D.P., J.P.C., A.S. and M.L.K. managed the mouse colonies. J.P., S.D.P., J.P.C., L.S. and

D.T.O. generated figures. A.P.A. and J.T. provided critical intellectual input and samples. J.P., S.D.P., J.P.C. and D.T.O. wrote the manuscript with input from A.P.A., P.G. and E.H. M.G., O.S., E.H. and DTO supervised the project. All authors had the opportunity to edit the manuscript and approve the final manuscript.

## Funding

## Competing interests
The authors declare no competing interests.
