## [Peer Review File · Nature Communications]

Four-Core Genotypes mice harbour a 3.2MB X-Y translocation that perturbs *Tlr7* dosageREVIEWER COMMENTS

Reviewer #1 (Remarks to the Author):

The manuscript by Panten et al. presents a comprehensive and timely investigation of a X-Y genetic translocation in the Four-core genotypes mouse model widely used to discriminate the function of sex chromosomes versus sex hormones in various pathophysiological contexts.

The authors clearly demonstrate that a 3.2 MB region of the X chromosome has translocated to the Sry-deficient Y chromosome, resulting in the increased expression of multiple genes, including Tlr7, a master regulator of the immune responses to viruses and systemic autoimmunity. Indeed, a similar translocation has been previously identified in the Yaa BXSB spontaneous model of systemic autoimmunity. The author provides a comprehensive list of publications using the FCG model and estimate that this genetic alteration has been circulating for at least a decade, suggesting that interpretation of these studies must be revisited.

This is a very important observation for the scientific community investigating the role of sex-specific factors in health and diseases.

I found the manuscript easy to read, it is clearly written and well supported by the data presented.

Reviewer #2 (Remarks to the Author):

This manuscript by Jasper Panten and colleagues makes an important observation that there is a duplicated region of the X chromosome which has become translocated onto the Y chromosome of the 4-Core genotypes mouse model system. The authors report that the YSry- chromosome contains a 3.2 MB region of the X chromosome encompassing ~9 genes including Tlr7. The data supporting this observation of a translocated region are rigorous and convincing, and this short manuscript would be appropriate for the 'Matters Arising' or brief report format. This manuscript reports a problem with the 4-core model, and the authors have also notified JAX and Dr. Art Arnold (who also distributes these mice to the research community investigating sex differences). The authors suggest that this translocation compromises the conclusions of ~90 publications involving these mice (Table 1), but the manuscript does not elaborate on this important point. In fact, one could argue that this translocation event may support the hypothesis that increased dosage of one/some of these 9 X-linked genes, independent of testosterone, is important in some systems/contexts examined in the ~90 publications using these mice. This is something that the authors should consider. In addition, the data in Supplemental Figure 6 suggests that this duplication does not significantly impact autosomal gene expression, which downplays the significance of this duplication event for impacting the interpretations of the ~90 publications in Table 1. This manuscript would strongly benefit from additional analyses that contextualize the observation of the translocation event for the field of sex differences and X-linked gene dosage, extending beyond this observation that 'there's a problem'.

Major comments:

1. Which of the 9 X-linked genes, besides Tlr7, has been demonstrated to exhibit a phenotype with increased gene dosage and/or increased gene expression? Similarly, what are the functions of the other 8 X-linked genes, which tissues/cells are they expressed in? Is there any information that would inform the interpretation of the outcomes of the studies in Table 1?
2. For Table 1, the authors should add additional information summarizing the major conclusions, the X-linked gene(s) within the 3.2MB region that is expressed in affect tissue/cells, and how the X-linked gene duplications/X-translocation might impact the finding(s). For example, if none of the 9 X-linked genes are expressed in brain, perhaps the major conclusions would not be impacted. Also for Table 1, the publications should be sorted by date for clarification, as JAX suggests that the translocation event arose around 2008. Along these lines, are there publications prior to 2008 whose data/conclusions are recapitulated in subsequent studies after 2008 (therefore suggesting

that the duplication may not have an impact on some tissues)?

3. Supplemental Figure 4: For 4a, the authors should correlate these observations from spleen with published studies of immune cell differences from mice. It looks like the data for NK cells is in agreement with known sex differences. In addition, the authors need to include wildtype controls for spleen derived immune cells as done for 4b for liver. This is important as it will reveal if the duplication impacts immune cell numbers.

Minor comments:

1. The description of Tlr7 as a 'auto-immune master regulator' is misleading, as its primary function is an endosomal toll like receptor family member that recognizes single-stranded RNA.

2. Figure 1: the figure legends are mislabeled for cell type. Also, are the genes (circles) just X-linked genes or all genes? Are all 9 genes identifiable/expressed for all 3 cell types? Is there a reason why liver cells & Kupffer cells were examined? Also, the colors and size of circles in Figure 1 are difficult to read.

3. Figure 2: the legend for Figure 2D is confusing because it's not clear whether the region is just Msl3 gene or additional genes encompassing a 200MB region of the X chromosome. The addition of a schematic for the F1 mouse model used for this study (the cross and resulting offspring) would be helpful for readers.

4. Supplemental Figure 2: For 2c, it's not clear why there are Y-linked reads from XX splenocytes. Also, what is the reason why the reads from splenocytes are different from liver reads. Do the authors have an explanation for why the total X-linked reads are higher in XY splenocytes (is this just from the 3.2MB translocation event)?

5. Supplemental Figure 5: the labeling of wildtype males is inconsistent with the labeling in other figures. The authors should also expand on the observations from this figure in the text itself.

6. Supplemental Figure 6: There is an incomplete sentence for figure title (in the figure legend). The colored dots are hard to see.

7. Supplemental Figure 7: The authors should re-order the immune cells so that they are listed in the same order, so it is easier for the reader to compare gene expression levels across cell types for each X-linked gene. Also, a correction with the text is necessary for lines 55-57, as the text says "immune cells in both organs show..." but this only shows spleen.

8. Supplemental Figure 8: there is a mistake with the label for 8c.

9. Supplemental Figure 9: the authors should label the autosomes beyond alternating colors.

**Responses to reviewer comments**

We thank the reviewers for their helpful feedback and comments which have been taken into consideration
in the detailed responses below, and which helped us in improving the manuscript. The primary focus of
our original submission was to highlight the duplication-translocation encountered in commercially
available FCG mice and the increased expression of the translocated genes. Our revision addresses most of
the points raised by the reviewers and includes two major changes not requested by the reviewers:

1) In our resubmission, our manuscript offers an alternative solution for the scientific community studying
sex differences. We now report that Dr. James Turner at the Crick Institute has maintained a separate FCG
colony, which does not carry the duplication-translocation (Supplementary Figures 14 and 15,
Supplementary Table 3). We have also determined that the mice submitted to Jax for distribution in 2010
indeed carried the translocation (Supplementary Figures 14 and 15, Supplementary Table 3).

2) We have added Nanopore and Targeted Locus Amplification datasets to confirm the translocation on the
Y^{Sry-} and narrow the precise junction point between the duplicated X region and the perturbed Y^{Sry-}
chromosome (Supplementary Figures 12 and 13).

**Reviewer #1 (Remarks to the Author):**

The manuscript by Panten et al. presents a comprehensive and timely investigation of a X-Y genetic
translocation in the Four-core genotypes mouse model widely used to discriminate the function of sex
chromosomes versus sex hormones in various pathophysiological contexts.

The authors clearly demonstrate that a 3.2 MB region of the X chromosome has translocated to the Sry-
deficient Y chromosome, resulting in the increased expression of multiple genes, including Tlr7, a master
regulator of the immune responses to viruses and systemic autoimmunity. Indeed, a similar translocation
has been previously identified in the Yaa BXSB spontaneous model of systemic autoimmunity. The
author provides a comprehensive list of publications using the FCG model and estimate that this genetic
alteration has been circulating for at least a decade, suggesting that interpretation of these studies must be
revisited. This is a very important observation for the scientific community investigating the role of sex-
specific factors in health and diseases. I found the manuscript easy to read, it is clearly written and well
supported by the data presented.

We thank the reviewer for their positive evaluation of our work.

**Reviewer #2 (Remarks to the Author):**

This manuscript by Jasper Panten and colleagues makes an important observation that there is a
duplicated region of the X chromosome which has become translocated onto the Y chromosome of the 4-
Core genotypes mouse model system. The authors report that the Y^{Sry-} chromosome contains a 3.2 MB
region of the X chromosome encompassing ~9 genes including Tlr7. The data supporting this observation
of a translocated region are rigorous and convincing, and this short manuscript would be appropriate for
the ‘Matters Arising’ or brief report format. This manuscript reports a problem with the 4-core model, and
the authors have also notified JAX and Dr. Art Arnold (who also distributes these mice to the research

community investigating sex differences). The authors suggest that this translocation compromises the
conclusions of ~90 publications involving these mice (Table 1), but the manuscript does not elaborate on
this important point. In fact, one could argue that this translocation event may support the hypothesis that
increased dosage of one/some of these 9 X-linked genes, independent of testosterone, is important in
some systems/contexts examined in the ~90 publications using these mice. This is something that the
authors should consider. In addition, the data in Supplemental Figure 6 suggests that this duplication does
not significantly impact autosomal gene expression, which downplays the significance of this duplication
event for impacting the interpretations of the ~90 publications in Table 1. This manuscript would strongly
benefit from additional analyses that contextualize the observation of the translocation event for the field
of sex differences and X-linked gene dosage, extending beyond this observation that ‘there’s a problem’.

Regarding the importance of the extra dosage of the 9 duplicated genes in different contexts/systems of the
90 publications of Table 1, we are reluctant to speculate too sharply on the severity of the impact this
translocation might have on any particular study. First, with Table 1 our goal is to highlight the studies that
could be impacted, so that others in the community can be mindful to look closely. Indeed, it is also possible
that some studies validated their FCG findings in complementary systems. Second, it is difficult to know
whether a particular study used a colony or strain of mice that did have the translocation, or not. Our
resubmission has now dated the translocation to the founding of the widely accessed Jax colony in 2010,
though we lack evidence of its earlier existence. Third, our dataset and context reveal a mild effect on
autosomal gene expression in the liver and spleen (per Supplementary Figure 6); other cell types and/or
tissues may have greater or lesser transcriptional perturbation and it will be difficult to rule out if phenotypic
and molecular differences might result from the extra dosage of one or more genes within the translocated
region. We discuss this third point further in Major Comment 2.

**Major comments:**

1. Which of the 9 X-linked genes, besides *Tlr7*, has been demonstrated to exhibit a phenotype with
increased gene dosage and/or increased gene expression?

Similarly, what are the functions of the other 8 X-linked genes, which tissues/cells are they expressed in?

Is there any information that would inform the interpretation of the outcomes of the studies in Table 1?

These are very pertinent questions and we thank the reviewer for bringing them up. We have added a new
Supplementary Table 2 summarizing published data related to the cell/tissue types where translocation
genes are expressed and the functional consequences of increased expression of the translocation genes.
The new table reveals two important points. First, four of the nine genes (*Hccs*, *Msl3*, *Prps2* and *Tmsb4x*)
encoding key regulators of cellular physiology are highly expressed in dozens of tissue contexts. Notably,
*Tmsb4x* is expressed in every cell type found in the Tabula Muris dataset. Second, for eight out of nine of
these genes, published functional data in a diversity of cell types shows that their overexpression has a
diversity of phenotypically and molecular consequences (NB: we could not find any overexpression studies
for *Hccs*).

2. For Table 1, the authors should add additional information summarizing the major conclusions, the X-
linked gene(s) within the 3.2MB region that is expressed in affect tissue/cells, and how the X-linked gene
duplications/X-translocation might impact the finding(s). For example, if none of the 9 X-linked genes are

expressed in brain, perhaps the major conclusions would not be impacted. Also for Table 1, the publications
should be sorted by date for clarification, as JAX suggests that the translocation event arose around 2008.
Along these lines, are there publications prior to 2008 whose data/conclusions are recapitulated in
subsequent studies after 2008 (therefore suggesting that the duplication may not have an impact on some
tissues)?

We thank the reviewer for these suggestions. Supplementary Table 1 is designed to be a neutral resource,
illustrating the potentially widespread implications of our findings. We strongly feel that we cannot
realistically pass such judgment on all the studies that have used FCG mice. We are suggesting careful
consideration of studies that use the FCG model. As indicated in the newly added header of
Supplementary Table 1 the listed studies may not be compromised by our finding as many of them might
dissect phenotypes unaffected by the translocation and/or validate results with orthogonal approaches.
Finally, the colonies used in the 90 studies might not necessarily carry the translocation, as we know
now that the MF1 strain doesn't have the translocation.

As shown in the new Supplementary Table 2 and described above for Reviewer 2, in Major Point 1, the
Tabula Muris dataset shows widespread expression of the 9 X-linked fusion genes that could directly - or
indirectly - impact any study using FCG mice carrying this translocation. For example, expression of *Tlr7*
and *Tmsb4x* in microglia (Supplementary Table 2) could influence FCG-based studies of the nervous
system and behaviour, which comprise over half of the studies published since 2008. Similarly, the
circulating immune cells, which express 7 out of these 9 genes, could influence the biology of any somatic
tissue.

Last but not least, we have sorted Supplementary Table 1 by year of publication. This change has improved
its useability.

3. Supplemental Figure 4: For 4a, the authors should correlate these observations from spleen with
published studies of immune cell differences from mice. It looks like the data for NK cells is in agreement
with known sex differences.

In addition, the authors need to include wildtype controls for spleen derived immune cells as done for 4b
for liver. This is important as it will reveal if the duplication impacts immune cell numbers.

Consistent with a recent publication (Cheng et al, 2023, Nature Immunology), we observed proportionally
more NK cells in gonadal male (XXT and XYT) spleens compared with their gonadal female counterparts
(XXO and XYO) (Supplementary Figure 4a). Additionally, we observed proportionally more NK cells in
XX spleens compared with their XY counterparts (Supplementary Figure 4a). Beyond NK cells, we
observed: proportionally more naive CD8 T cells (consistent with Ghosh et al [2021]), gamma delta T cells
and regulatory T cells (contradicting Gandhi et al [2022]) in gonadal female spleens; proportionally more
transitional B cells in gonadal male spleens; and more basophils in XX spleens (Supplementary Figure 4a).

The reviewer suggests the inclusion of wild-type mice for spleen analysis, which would have allowed us to
evaluate how this genome duplication impacts immune cell numbers in multiple tissues. Our discovery of
this X-Y translocation was fortuitous, and was based on quite different datasets from spleen (scRNA-seq)

and liver (single-nucleus RNA-seq). To add wild-type males to spleen FCG data would demand that we
repeat all the experiments at once, in order to avoid any batch effects. Unfortunately, this experiment is not
possible, because we have replaced this colony with FCG mice that do not carry the translocation.

**Minor comments:**

1. The description of Tlr7 as a 'auto-immune master regulator' is misleading, as its primary function is an
endosomal toll like receptor family member that recognizes single-stranded RNA.

To reflect that Tlr7's function goes beyond a critical role in autoimmune disease pathogenesis (Pisitkun et
al, 2006; Brown et al, 2022), we have replaced "auto-immune master regulator" in the abstract (line 23) and
main text (lines 42-43).

2. Figure 1: the figure legends are mislabeled for cell type.
Also, the colors and size of circles in Figure 1 are difficult to read.
Also, are the genes (circles) just X-linked genes or all genes?

We have modified the legend text to reflect the panel of Figure 1. We increased the size of the dots
representing the translocated genes statistically different between the two compared conditions. The
analysis presented in Figure 1 specifically focuses on X-chromosome genes, and this information has been
incorporated into the legend text of Figure 1.

Are all 9 genes identifiable/expressed for all 3 cell types?

We have added the following to the main text (lines 56-59): "The identified set of nine X-linked genes
showed cell type-specificity in their level of expression. Five of the nine genes (*Arhgap6*, *Msl3*, *Frmpd4*,
*Prps2* and *Tmsb4x*) are detectably expressed in splenocytes, hepatocytes and Kupffer cells; however, *Hccs*
and *Amelx* are not expressed in Kupffer cells and *Tlr7* and *Tlr8* are not expressed in hepatocytes (Figure
1c-h; Supplementary Figure 6)."

Is there a reason why liver cells & Kupffer cells were examined?

Hepatocytes were examined because they are the most abundant cell type of the liver and consequently
well-represented in our sci-RNA-seq dataset (Supplementary Figure 3). Kupffer cells were examined
because they are the most abundant immune cell type of the liver, and we wanted to evaluate the impact of
the translocation on immune cells in particular because of Tlr7's presence in the translocation and well-
established functional importance in immune contexts.

3. Figure 2: the legend for Figure 2D is confusing because it's not clear whether the region is just Msl3
gene or additional genes encompassing a 200MB region of the X chromosome.

As shown in the UCSC genome browser screenshot below, the bacterial artificial chromosome (BAC) probe
used (RP23-391N16, depicted in green in the below panel) in our DNA FISH experiments targets a ~200kb
region on the X chromosome containing the complete *Ms13* gene and a small part of adjacent *Frmpd4* (exon
1 and partial intron 1). We have updated the main text (lines 91-94) and the legend text of Figure 2 to
enhance clarity.

The addition of a schematic for the F1 mouse model used for this study (the cross and resulting offspring)
would be helpful for readers.

We have added a cartoon to guide readers through the results (panel c of the revised Figure 2).

4. Supplemental Figure 2: For 2c, it's not clear why there are Y-linked reads from XX splenocytes.

In the figure below we show that a single Y-linked gene (*Gm47283*, also known as *Erdr1*, depicted in red
in the below panel) is expressed at approximately the same level in XX and XY splenocytes independent
of gonad type. *Gm47283* is present in the pseudoautosomal region (PAR) of the Y chromosome and a
second homologous gene is present in the PAR of the X chromosome (*Gm21887*). Therefore, we infer that
reads from X-linked *Gm21887* were misassigned to Y-linked *Gm47283*.

Offtarget mapping to the Y

Also, what is the reason why the reads from splenocytes are different from liver reads.

Genes beginning with the Gm- prefix (predicted genes and/or pseudogenes) were filtered out during
processing of the liver snRNA-seq data but not the spleen scRNA-seq data. We have performed such
filtering on the spleen dataset and updated Supplementary Figure 2c.

Do the authors have an explanation for why the total X-linked reads are higher in XY splenocytes (is this
just from the 3.2MB translocation event?)?

We believe this is mainly due to *Tmsb4x*, which has extremely high read counts specifically in the spleen
scRNA-seq data set. Indeed, removing this gene from the calculation eliminates this effect (see the second
row of the below figure). Overall the immune cells have a higher expression of this translocated gene. In
liver the X chromosome counts are comparable between XX and XY, presumably because of proportionally
fewer immune cells highly expressing the translocated gene. We added this explanation to the legend text
of Supplementary Figure 2.

5. Supplemental Figure 5: the labeling of wildtype males is inconsistent with the labeling in other figures.

We changed the wild-type label in Supplementary Figure 5 in XYT-WT and adjusted the plots of the entire
manuscript to maintain consistent labelling across all figures.

The authors should also expand on the observations from this figure in the text itself.

For panel A, the presence of the XYT-WT in the liver dataset enabled us to cleanly and unambiguously
show the elevated expression of these X-linked genes in Kupffer cells and hepatocytes (lines 50-51).

We have deleted panel B, which compared XYO versus XYT-WT. In hindsight, this comparison provided
a confusing message at best, because there were two variables changing at once: gonad and translocation.
We believe that the removal of this panel improves the clarity of our message.

6. Supplemental Figure 6: There is an incomplete sentence for figure title (in the figure legend). The colored
dots are hard to see.

We have modified the legend text as follows “The increased expression of the translocated genes has a
minor effect on autosomal gene expression.” and made the coloured dots, representing genes with 2-fold
change and an adjusted p-value < 0.1, together with their annotation bigger.

7. Supplemental Figure 7: The authors should re-order the immune cells so that they are listed in the same
order, so it is easier for the reader to compare gene expression levels across cell types for each X-linked
gene.

We re-order the cells so that they are listed in the same order for all the genes.

Also, a correction with the text is necessary for lines 55-57, as the text says “immune cells in both organs
show...” but this only shows spleen.

We changed the sentence and made it a more precise explanation of where (in which cells) the 9 genes are
expressed in our dataset. See lines 56 onwards: “The identified set of nine X-linked genes showed cell type-
specificity in their level of expression. Five of the nine genes (*Arhgap6*, *Msl3*, *Frmpd4*, *Prps2* and *Tmsb4x*)
are detectably expressed in splenocytes, hepatocytes and Kupffer cells; however, *Hccs* and *Amelx* are not
expressed in Kupffer cells and *Tlr7* and *Tlr8* are not expressed in hepatocytes (Figure 1c-h; Supplementary
Figure 6)”

8. Supplemental Figure 8: there is a mistake with the label for 8c.

We changed the label of Supplemental Figure 8c to “duplicated”.

9. Supplemental Figure 9: the authors should label the autosomes beyond alternating colors.

In addition to alternating colours, we have added the chromosome number in blue for each panel in both
Supplemental Figure 9 and Supplemental Figure 14.

REVIEWERS' COMMENTS

Reviewer #1 (Remarks to the Author):

As mentioned in prior reviews, this manuscript will be a very useful tool and data set for the immunology community. It carries important genetic information regarding the FCG mouse model, widely used by the scientific community studying sex differences.

Within this revision, the authors have addressed all concerns raised by the reviewers and provided two major changes not requested which greatly improved the manuscript and its general broad interest.

Reviewer #2 (Remarks to the Author):

The authors have revised the manuscript and now has increased the readability and interpretation of the results. The addition of a new table for the duplicated genes that specifies their functions, whether any phenotype was observed with increased expression, and the cell types expressing that gene (which is useful).

There are some minor edits that are still required:

- Abstract has an awkward sentence "We report that a copy of a 3.2 MB region of the X chromosome has translocated to the YSry- chromosome and thus increased the expression of 'X genes' ..." — I think "-linked" is missing.
- Figure 1 Legend includes "c-f" as citing differential expression from all spleen cells, but only panels c and f show the spleen (they are using "-" as a comma).

Responses to reviewer comments

We thank the reviewer for their positive evaluation of our work and for their comments which have been taken into consideration in the detailed responses below.

Reviewer #1 (Remarks to the Author):

As mentioned in prior reviews, this manuscript will be a very useful tool and data set for the immunology community. It carries important genetic information regarding the FCG mouse model, widely used by the scientific community studying sex differences.

Within this revision, the authors have addressed all concerns raised by the reviewers and provided two major changes not requested which greatly improved the manuscript and its general broad interest.

Reviewer #2 (Remarks to the Author):

The authors have revised the manuscript and now has increased the readability and interpretation of the results. The addition of a new table for the duplicated genes that specifies their functions, whether any phenotype was observed with increased expression, and the cell types expressing that gene (which is useful).

There are some minor edits that are still required:

- Abstract has an awkward sentence “We report that a copy of a 3.2 MB region of the X chromosome has translocated to the Y^{Sry}- chromosome and thus increased the expression of ‘X genes’ ...” — I think “-linked” is missing.

Line 18 – 19: we added, We report that a copy of a 3.2 MB region of the X chromosome has translocated to the Y^{Sry}- chromosome and thus increased the expression of X-linked genes

- Figure 1 Legend includes “c-f” as citing differential expression from all spleen cells, but only panels c and f show the spleen (they are using “-” as a comma).

Line 405-406: we added, **c, f** Differential expression analysis of X chromosome genes